# Epidemiological Characteristics of Road Traffic Injuries Involving Children in Three Central American Countries, 2012–2015

**DOI:** 10.3390/ijerph18010037

**Published:** 2020-12-23

**Authors:** Virginia Núñez-Samudio, Francisco Mayorga-Marín, Humberto López Castillo, Iván Landires

**Affiliations:** 1Instituto de Ciencias Médicas, Las Tablas, Los Santos 0701, Panama; vnunez@institutodecienciasmedicas.org; 2Sección de Epidemiología, Departamento de Salud Pública, Región de Salud de Herrera, Ministry of Health, Chitré, Herrera 0601, Panama; 3Centro de Investigaciones y Estudios de la Salud, Universidad Nacional Autónoma, Managua 10000, Nicaragua; francisco.mayorga@cies.unan.edu.ni; 4Department of Health Sciences, College of Health Professions and Sciences, University of Central Florida (UCF), Orlando, FL 32816, USA; humberto.lopezcastillo@ucf.edu; 5Department of Population Health Sciences, College of Medicine, UCF, Orlando, FL 32827, USA; 6Centro Regional Universitario de Azuero (CRUA), Universidad de Panamá, Chitré, Herrera 0601, Panama; 7Hospital Joaquín Pablo Franco Sayas, Región de Salud de Los Santos, Ministry of Health, Las Tablas, Los Santos 0701, Panama

**Keywords:** childhood, injury prevention, infant mortality, traffic crashes, Central America

## Abstract

Although motor vehicle collisions (MVCs) are a worldwide public health concern due to their high injury, mortality, and fatality rates, few studies have addressed the epidemiologic behavior of MVCs in Latin American youth. Thus, this study was aimed at describing and comparing the characteristics of MVCs involving 0 to 14-year-olds in Costa Rica, Guatemala, and Panama. A secondary aim was to estimate the crude MVC-related injury, fatality, and mortality rates and their trends over time. We conducted a descriptive, retrospective study using publicly available data for Costa Rica, Panama, and Guatemala between 2012 and 2015. We examined the reported MVC cases and calculated the crude injury, fatality, and mortality rates and their trends over time (α = 0.05). Publicly available data reported 12,020 MVC-related injuries and 431 MVC-related deaths involving 0 to 14-year-olds. The most frequent mechanisms involved 0 to 14-year-olds as passengers or pedestrians in MVCs (>85% of all cases). The highest crude MVC-related injury and mortality rates were reported for Panama (119.35 and 2.14 per 100,000 population, respectively, in 0 to 14-years-olds), while Guatemala had the highest median MVC-related fatality rate (8.84 per 100,000 events; χ^2^ [2] = 377.8; *p* < 0.001) with a statistically significant trend increasing over time (*r* = 0.947; *p* = 0.027). Although several factors play a role in the prevention of MVCs among 0 to 14-year-olds, we found that Costa Rica was the only country that implemented a policy on child restraint systems resulting in the lowest rates of MVC-related injury, mortality, and fatality. These results could be used by decision makers from the aforementioned Central American countries to develop adequate policies addressing MVC preventative strategies to protect Central American infants and children.

## 1. Introduction

Motor vehicle collisions (MVCs) account for 30% of all injury-related deaths before the age 20 [1]. Using worldwide MVC data as of 2018, it was estimated that MVCs cause 1.35 million deaths annually and are the key cause of mortality among 5- to 14-year-olds [1]. In Latin America and the Caribbean, where almost half of the population is under 25, the mortality rate related to MVCs among 5- to 14-year-olds is almost twice the world’s mean rate [2]. In 2014, Al-Madani [3] reported data on MVCs for North and Central America, where 29 countries accounted for 77,390 of the MVC fatalities (9.4% of global events). The trend of deaths in these countries shows a significant decrease of 24% over time; however, when the United States (US) and Canada are removed from the model, the remaining 27 countries show an increase in the trend of MVC fatalities [3].

Few studies have reported MVC fatalities focusing on under-20-year-olds. In 2017, this age group represented 16.0% of the 717 MVC fatalities reported in Costa Rica that year [4]. Although there were no specific subanalyses by age in the study, most fatalities occurred in March and April (20.5%), on Saturdays and Sundays (41.6%), at night (25.0%), at the site of the MVC (55.8%), and among males (83.3%) [4]. In the same year, a report in Belize estimated that 61 MVC fatalities accounted for 2501 years of potential life lost, along with 338 hospitalizations and 565 outpatient injuries. Moreover, the economic cost of MVCs was just over 11 million USD (0.9% of Belize’s GDP) [5]. A more recent study conducted in 2016 in Panama, a country with few traffic safety regulations for infants and children, showed that MVC-related infant and childhood mortality rate was four times higher than that of Spain, a country with stricter regulations on childhood retention systems (CRSs) [6].

Child retention systems, which comprise safety seats for infants and younger children and booster seats for older children, constitute the most critical individual-level intervention to reduce MVC fatalities among infants and children [7]. Because CRS use depends on implementation by caregiving adults, interventions educating caregivers have demonstrated a reduction of almost 50% in the number of children not riding in a CRS [8]. Correct implementation and use of CRSs in vehicles reduces MVC mortality by 71% in under-1-year-olds, by between 54% and 80% in 1- to 4-year-olds [5,6,7], and by 19% in 8- to 12-year-olds [9]. However, although there are international consensus statements on the use of CRSs [1,2,10], local regulations for their use vary widely among Latin American countries. In turn, the variation in the annual infant and childhood MVC-related mortality rates in Latin American countries, directly and indirectly, reflects the CRS implementation strategies implemented by each country [2,10,11].

Even though MVC-related injury continues to be an important public health problem worldwide, there is a dearth of studies on MVC-related mortality and fatality rates in Central America that could potentially inform the need for preventative measures, such as CRSs. Thus, this study aimed to describe and compare the characteristics of MVCs involving 0 to 14-year-olds in Costa Rica, Guatemala, and Panama. A secondary aim was to estimate the crude MVC-related injury, fatality, and mortality rates in these three Central American countries and their trends over time.

## 2. Methods

We conducted a descriptive, retrospective study to understand the epidemiology of traffic-related infant and childhood mortality in Costa Rica, Panama, and Guatemala between 2012 and 2015. These three countries were selected because they have official online reports of MVC-related injuries by age for the study period, and combined, these three countries represent 45% of the area and 55% of the population of Central America. Table 1 presents an overview comparing the three countries.

### 2.1. Data Sources

Cases of MVC-related injury and death among 0 to 14-year-olds from 2012 to 2015 were available at online databases and were downloaded from Costa Rica’s *Consejo de Seguridad Vial* [12], *Guatemala’s Departamento de Tránsito de la Policía Nacional Civil* [13], and Panama’s *Instituto Nacional de Estadística y Censo* [14].

The datasets also specified the mechanisms of MVC, although the reporting was not standard. To facilitate comparisons, we classified the mechanisms of MVC reported by each country in one of the following four categories:Passenger in an MVC: the victim was a passenger in the MVC. This category excludes vehicles moved solely by human power, motorized wheelchairs, and motorcycles.Pedestrian in an MVC: the victim was along a road where the MVC occurred.Rollover in an MVC: the victim was in an overturned motor vehicle.Other: the victim fell from a motor vehicle in motion, the victim presents a combination of the previous categories, or the mechanism of the MVC is not otherwise classified.

To estimate the crude injury and mortality rates, we obtained the total population estimates for 0 to 14-year-olds in each country and in each year from The World Bank [15].

### 2.2. Data Analyses

Frequencies. We extracted the datasets and compared the absolute and relative frequencies of the total number of victims, injuries, deaths, and mechanisms of MVC reported by the three countries for 0 to 14-year-olds from 2012 to 2015. Next, we calculated three different crude rates, understood as the total number of events divided by the population at risk in the same period and in the same geographic area, multiplied by a constant, which is a multiple of 10. Medians reported for these crude rates were calculated by averaging the two middle values in the four-point data series.

Crude injury rate: we calculated the crude injury rate (per 100,000 population 0 to 14 years-old) by dividing the total number of injuries with the respective year’s estimate for the total population 0 to 14-years-old.

Crude mortality rate: we calculated the crude mortality rate (per 100,000 population 0 to 14 years-old) by dividing the total number of deaths with the respective year’s estimate for the total population 0 to 14-years-old.

Crude fatality rate: we calculated the crude fatality rate (per 100 events) by dividing the total number of deaths with the number of MVC events reported in each country each year. Fatality rates were compared with an independent sample χ^2^ test, setting alpha at 0.05.

Trends over time. We used ordinary least squares (OLS) linear regression to estimate the trends over time, with overall correlation (*r*) and determination (*R*^2^) coefficients set at the 0.05 alpha level.

## 3. Results

Costa Rica, Guatemala, and Panama were the only Central American countries with comparable, publicly available, official data during the study period. Of note, none of the countries included in the study had a legal mechanism for obtaining a driver’s permit or license before 16 years of age. We found a total of 12,020 MVC-related injuries and 431 MVC-related deaths involving 0 to 14-year-olds. Table 2 details these frequencies by year and country.

The three countries had different reporting categories for traffic events (Figure 1). The most frequent mechanism of MVC in Costa Rica (58%) and Guatemala (49%) was MVC as a passenger. In Panama, the most frequent mechanism of MVC was MVC as a pedestrian (43%), followed closely by MVC as passenger (42%). Together, rollovers and other MVCs represented less than 20% of all reported MVC-related deaths.

The median, annual, crude, MVC-related injury rates (per 100,000 population, 0 to 14-years-old) were 80.85 for Costa Rica, 14.93 for Guatemala, and 119.35 for Panama. The median, annual, crude, MVC-related mortality rates (per 100,000 population, 0 to 14-years-old) for the study period were 0.87 for Costa Rica, 1.22 for Guatemala, and 2.14 for Panama. The overall annual MVC-related fatality rates were 1.08 for Costa Rica, 8.84 for Guatemala, and 1.83 for Panama (χ^2^ [2] = 377.8; *p* < 0.001). Table 3 presents the crude injury, mortality, and fatality rates by year for each country.

Trends over time analyses by country (Table 4) revealed that in Panama, all three rates of mortality, injury, and fatality showed a decreasing trend. However, these trends did not reach statistical significance. Costa Rica showed the same downward trend for mortality and fatality rates, but the injury rates tended to increase by 37.6% each year, again, without reaching statistical significance. In the case of Guatemala, however, all three rates of mortality, injury, and fatality increased to a statistically significant extent (*p* < 0.05) every year by at least 75.6%.

## 4. Discussion

To the best of our knowledge, this study is the first one to use publicly available, official country-level data to describe, compare, and estimate rates of MVC-related injuries in three Central American countries. We found that in the four years of the project, 12,020 infants were involved in an MVC, which resulted in 431 deaths. Victims involved in MVCs as a passenger or as a pedestrian represented 80% or more of all mechanisms of MVC. A recent systematic review found that child pedestrian safety is affected by three main factors (i.e., built environment, drivers, and vehicles) and four cross-cutting critical issues (i.e., reliable collision and exposure data, evaluation of interventions, evidence-based policy, and intersectoral collaboration) [16]. To this end, the WHO recommends the Safe System Approach, which considers people’s vulnerability to serious injuries through a system designed to be more accepting of human error [17].

Although crude MVC-related injury rates were higher in Panama and Costa Rica, Guatemala demonstrated a statistically significant trend towards increases in its injury rates. On the other hand, Panama and Guatemala presented the largest crude MVC-related mortality rates. However, trend over-time analyses revealed that in Panama there was a trend towards decreased rates over time, while in contrast, in Guatemala crude mortality rates increased significantly. The case fatality rates are higher in Guatemala, with a statistically significant trend to increase over time.

In Latin America, traffic events are responsible for 1.43% of all deaths among infants and children under 5-years old and 13.83% of all deaths among children 5- to 14-years old [17]. Our data showed that MVC-related mortality rates tend to decrease over time for Costa Rica and Panama but not for Guatemala. A previous study on infant and childhood traffic safety in Panama demonstrated that traffic-related deaths in infants and children represent a significant public health issue, with the highest number of deaths and fatality rates reported for children younger than five years. The mean annual MVC-related mortality rate (per million population) in 0 to 14-year-olds is 6 worldwide, 6 in the European Union, and 32 in Latin America [11]. Our data showed higher mean mortality rates for comparable populations in Panama (3.7 times higher), Guatemala (2.2 times higher), and Costa Rica (1.3 times higher). Thus, Costa Rican statistics seem closer to the worldwide and European means.

MVC-related fatality rates are reflective of both the primary prevention strategies deployed by countries to avoid MVCs altogether and the secondary prevention measures in place to respond and mitigate MVCs when they occur. We found that, while MVC-related fatality rates in Panama and Costa Rica tended to decrease, Guatemala showed a statistically significant trend upwards. These results should prompt an analysis of primary and secondary preventative measures in place to prevent and mitigate MVCs.

The United Nations’ Sustainable Development Goals [18] include an action plan for infant, childhood, and adolescent health. This plan confirms that there is a great challenge for the health agenda in the Central American region in terms of policies and resources to improve and support a safer built environment, safer drivers, and safer vehicles. The Central American Traffic Safety Manual is a concerted effort among the six Spanish-speaking countries to establish standardized, clear, and specific regulations on traffic safety [19]. This manual represents an opportunity for the countries in the region to develop, implement, and harmonize public policy on traffic safety for Central America. However, as of the time this manuscript was finalized, Costa Rica was the only country in the region enacting traffic-related policies in compliance with the manual and other international standards.

A specific preventative intervention in pediatric MVC-related injuries is the implementation of CRSs. Studies have demonstrated that the correct use of CRSs reduces mortality and injury frequency and severity in infants and children and consequently reduces injury-related sequelae and disability in this vulnerable group [7,8,9,16,20]. While Costa Rica has implemented legislation mandating the use of CRSs [21], Panama and Guatemala, lack specific legislation for CRS in compliance with international standards. Moreover, there are no education and awareness campaigns disseminating information about CRSs in Panama and Guatemala. In Panama, our group joined a private citizen effort to present a bill for traffic regulations for CRSs for 0- to 12-year-olds [22]. We did not find technical requirements for CRSs in Guatemala’s traffic regulation.

It has been documented that the proper installation and use of child restraint systems in vehicles reduced mortality in approximately 71% of passengers under one year of age and between 54 and 80% of passengers aged 1–4 [7,23,24]. Furthermore, seat belt use decreased mortality by 55–75% [25]. In a previous study, we demonstrated that the infant mortality rate in Panama from motor vehicle crashes is 3 to 4 times higher than the rates observed in a country with better child restraint legislation (Spain) [6]. That study concluded that the lack of specific legislation on the use of CRS (car seats), as well as the lack of information and awareness campaigns, could be responsible for the high number of child victims associated with motor vehicle collisions [6]. We have also documented our work and achievements in Panama through the generation of public policies on child road safety based on scientific evidence [26].

Although innovative, this study is not void of limitations. We used and matched multiple country-level datasets from various data sources to the best of our abilities. However, local differences embedded in each country’s reporting system may have influenced the accuracy of the data reported, and consequently, the precision of the rates estimated. Additionally, because the MVC-related data for 0-to-14-year-olds were reported as an aggregate, it was not possible to conduct subanalyses per age subgroups and to estimate age-adjusted rates for more adequate comparisons between countries. Because crude rates are influenced by the underlying age distribution of the country’s population [27], our estimates should be interpreted with this caution in mind. The limited number of four data pairs in the time series limits the sensitivity and power of the trend-over-time analyses. Our initial attempt was to obtain data from all Central American countries, but only the three countries included had relatively recent data for our analytic purposes. More up-to-date data (e.g., 2016–2020) were not yet available from the local government sources used in this study. However, we still consider the data and results relevant and current, but should be interpreted with caution in light of new data. Limitations aside, our results shed new light on the marked disparities in the MVC-related injury, mortality, and fatality rates among 0 to 14-year-olds in three Central American countries.

## 5. Conclusions

We found higher mortality and fatality rates for comparable populations in Panama and Guatemala than in Costa Rica. These results could be due to Costa Rica being the only country in the region with traffic safety policy enacted in compliance with international standards.

The large number of deaths in Panama and Guatemala and the high fatality rate in Guatemala could be the result of nonspecific legislation mandating CRSs and a lack of road safety preventative campaigns. The increase in fatality among under-5-year-olds is associated, at least partially, with a lack of specific legislation on retention systems and the lack of adequate information and education campaigns [26].

In Central America, public policies aimed at reducing MVC-related infant and childhood injury, mortality, and fatality rates urgently need to be created. This study provided evidence that MVCs involving children as passengers or pedestrians were the traffic events that accounted for the most fatalities. Policymakers could use this and other scientific evidence available to develop and implement policies and specific legislation on road safety and CRSs in compliance with sustainable development goals and decrease the traffic-related infant and childhood mortality rates.

## Figures and Tables

**Figure 1 ijerph-18-00037-f001:**
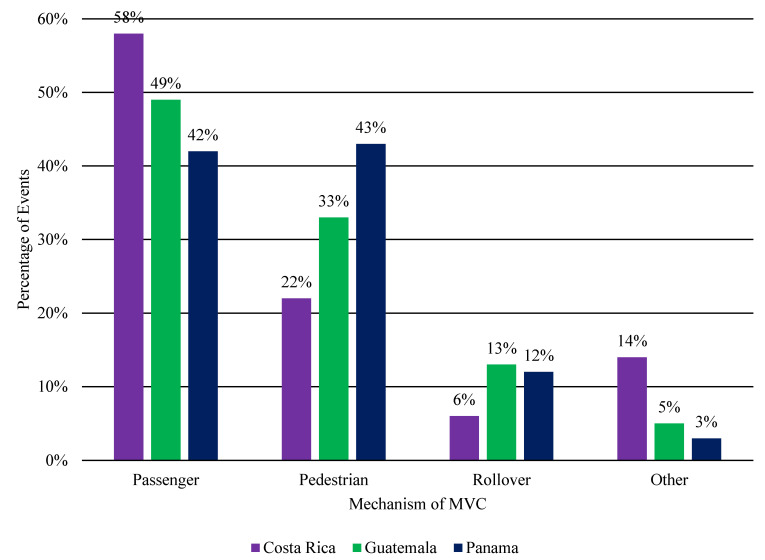
Most frequent type of MVCs in Costa Rica, Guatemala, and Panama 2012–2015. Abbreviation: MVC, motor vehicle collision.

**Table 1 ijerph-18-00037-t001:** Overview of the three countries included in the study.

Characteristic	Costa Rica	Guatemala	Panama
Area, km^2^	51,100	108,889	75,417
2018 population estimate, millions	5.00	17.26	4.18
Population density, km^−2^	84.9	129.0	56.0
Proportion of White/Mestizo, %	83.6	56.0	65.0
Minimum age for a driving permit, y	—	16	16
Minimum age for a driving license, y	18	18	18
2018 HDI	0.79	0.65	0.80
2019 Gini coefficient	47.8	48.3	49.9
2020 GDP (PPP) per capita, USD	12,690	8,413	28,456
Economy	Upper middle	Upper middle	High

Abbreviations: HDI, human development index; GDP, gross development product; PPP, purchasing power parity; USD, United States dollar; WHO, World Health Organization; y, years. Data sources: Wikipedia (www.wikipedia.org), The World Bank (databank.worldbank.org), and the WHO Data Platform (www.who.int/data).

**Table 2 ijerph-18-00037-t002:** Frequency of MVC-related injuries and deaths involving 0 to 14-year-olds in Costa Rica, Guatemala, and Panama 2012–2015.

Year	Costa Rica		Panama		Guatemala		Total	
	Injuries	Deaths	Injuries	Deaths	Injuries	Deaths	Injuries	Deaths
**2012**	759	10	1416	28	698	45	2873	83
**2013**	920	9	1283	25	607	39	2810	73
**2014**	869	10	1196	21	985	92	3050	123
**2015**	872	8	1335	22	1080	122	3287	152
**Total**	3420	37	5230	96	3370	298	12 020	431

Abbreviation: MVC, motor vehicle collision.

**Table 3 ijerph-18-00037-t003:** Crude MVC-related mortality, injury, and fatality rates for 0 to 14-year-olds in Costa Rica, Guatemala, and Panama 2012–2015.

Year	Costa Rica			Panama			Guatemala		
	CMR	CIR	CFR	CMR	CIR	CFR	CMR	CIR	CFR
2012	0.91	69.27	1.32	2.59	131.09	1.98	0.80	12.41	6.45
2013	0.83	84.63	0.98	2.30	117.87	1.95	0.69	10.77	6.43
2014	0.93	80.49	1.15	1.91	109.06	1.76	1.63	17.45	9.34
2015	0.75	81.21	0.92	1.99	120.83	1.65	2.16	19.15	11.30

Abbreviations: CFR, crude fatality rate per 100 events; CIR, crude injury rate per 100,000 population ages 0 to 14 years; CMR, crude mortality rate per 100,000 population ages 0 to 14 years; MVC, motor vehicle collision.

**Table 4 ijerph-18-00037-t004:** Trend-over-time analyses for MVC-related mortality, injury, and fatality rates for 0 to 14-year-olds in Costa Rica, Guatemala, and Panama 2012–2015.

Rate	Costa Rica			Panama			Guatemala		
	*r*	*R* ^2^	*p*	*r*	*R* ^2^	*p*	*r*	*R* ^2^	*p*
Mortality	−0.621	0.386	0.379	−0.910	0.829	0.421	0.927	0.859	0.010
Injury	0.613	0.376	0.495	−0.563	0.317	0.161	0.870	0.757	0.015
Fatality	−0.735	0.541	0.132	−0.968	0.937	0.523	0.947	0.898	0.027

## Data Availability

Publicly available datasets were analyzed in this study. These data were retrieved on 9 November 2020 from references [12,13,14,15].

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
