# Peer review of "Epidemiological Characteristics of Road Traffic Injuries Involving Children in Three Central American Countries, 2012–2015"

_ijerph, 2020, doi:10.3390/ijerph18010037_

Round 1

Reviewer 1 Report

The aim of the presented study is to describe and compare the characteristics of MVCs involving 0- to 14-year – olds in Costa Rica, Guatemala, and Panama. The article has the great merit of proposing a method of analysis with secondary data sources that brings agility and economy in obtaining the data. It is an interesting analysis from a technical point of view, but without elements that characterize its relevance in the scope of scientific knowledge on the subject. The relationship between the data studied and the absence of legislation on childhood retention systems is a promising hypothesis, but not proven in this study.

Some specific comments:

I suggest reviewing format aspects since there is missing space at the beginning of the sentence on lines 29 and 35.

I also suggest defining some nomenclatures that I believe are common in the medical field but not so common in the engineering area such as "mechanisms" and "crude"

Please include HDI in the list of abbreviations in Table 1.

The website addresses provided on lines 96, 97, and 98 of the data used in the study do not refer to specific pages of the data used, making it difficult to accurately locate the information used in the study. It would be interesting to provide a more specific link.

What do you mean by “unadjusted mortality rate”?

How were median, annual, crude rates obtained? Shouldn't it be the average of the 4 years shown in table 3? If so, the CIR of Costa Rica would be 78.9 and not 80.85. I find it useful to explain and include it in Table 3.

I also question the validity of the comments on lines 156 to 159 since these trends are not statistically significant. In such cases, it would not be possible to state the existence of these trends.

The discussions section brings elements not supported in the previous sections of the article (child pedestrian safety factors and Safe System Approach). This should not occur.

The statement presented in the discussion that “Panama and Costa Rica has a trend to decrease mortality rates over time” is not statistically proven in the article.

Author Response

Reviewer 1

The aim of the presented study is to describe and compare the characteristics of MVCs involving 0- to 14-year – olds in Costa Rica, Guatemala, and Panama. The article has the great merit of proposing a method of analysis with secondary data sources that brings agility and economy in obtaining the data. It is an interesting analysis from a technical point of view, but without elements that characterize its relevance in the scope of scientific knowledge on the subject. The relationship between the data studied and the absence of legislation on childhood retention systems is a promising hypothesis, but not proven in this study.

Response: We would like to thank reviewer 1 for his excellent review work which has contributed significantly to improving our manuscript. We are excited that reviewer 1 considers our study an interesting analysis.

It has been documented that the proper installation and use of child restraint systems in the vehicle reduced approximately 71% of mortality among passengers under one year of age and between 54 and 80% among passengers aged 1-4 (Desapriya et al., 2004; Durbin et al., 2005; Sauber-Schatz et al., 2014). Seat belt use has been shown to decrease mortality by 55-75% (Zhu et al. 2007).

In a previous study we have shown that the infant mortality rate in Panama from motor vehicle crashes is 3 to 4 times higher than the rates observed in a country with better child restraint legislation such as Spain. That study concluded that in Panama, the lack of specific legislation on the use of child restraint systems (car seats), as well as the lack of information and awareness campaigns, could be responsible for the high number of child victims associated with motor vehicle collisions (Núñez-Samudio V, Jaramillo-Morales J, Landires I. Prevalence and characteristics of child victims in motor vehicle collisions in Panama. Traffic Inj Prev. 2016 May 18;17(4):391-3 ). We have also documented our work and achievements in Panama through the generation of public policies on child road safety based on scientific evidence (Núñez-Samudio V, Landires I. Public policies based on scientific evidence: child road safety. Arch Argent Pediatr. 2020;118(3):e252-e257. doi:10.5546/aap.2020.eng.e252 )

Some specific comments:

I suggest reviewing format aspects since there is missing space at the beginning of the sentence on lines 29 and 35.

Response: Formatting was reviewed, and the spacing issues were corrected

I also suggest defining some nomenclatures that I believe are common in the medical field but not so common in the engineering area such as "mechanisms" and "crude"

Response: We appreciate the opportunity to expand the readership to non-medical audiences. In medicine, “mechanism” refers to the underlying processes driving diseases. On the other hand, “crude” or “unadjusted” rates refer to the raw calculation of the numerator divided by the denominator for each measure, without taking into account the specific composition of the population (commonly by age or sex). These are standard epidemiological methods that have been defined in the Methods and in the Limitations sections.

Please include HDI in the list of abbreviations in Table 1.

Response: Human Development Index (HDI) was added to the list of abbreviations in Table 1.

The website addresses provided on lines 96, 97, and 98 of the data used in the study do not refer to specific pages of the data used, making it difficult to accurately locate the information used in the study. It would be interesting to provide a more specific link.

Response: Specific links for each country have been added, namely:

  • Panama: https://www.inec.gob.pa/publicaciones/Default3.aspx?ID_PUBLICACION=555&ID_CATEGORIA=5&ID_SUBCATEGORIA=40
  • Costa Rica: http://repositorio.mopt.go.cr:8080/xmlui/bitstream/handle/123456789/3898/629.2-1.pdf?sequence=1&isAllowed=y
  • Guatemala: https://www.ine.gob.gt/estadisticasine/index.php/usuario/hechos_transito_menu

What do you mean by “unadjusted mortality rate”?

Response: A crude or unadjusted death rate is simply the number of deaths divided by the population at risk, often multiplied by some constant so that the result is not a fraction. These are standard calculations in epidemiology, but we understand a broader audience may have access to the journal and we have incorporated definitions in the Methods section.

How were median, annual, crude rates obtained? Shouldn't it be the average of the 4 years shown in table 3? If so, the CIR of Costa Rica would be 78.9 and not 80.85. I find it useful to explain and include it in Table 3.

Response: The median is the value that splits a dataset in two equal halves and is akin to the 50th percentile. In an even-number dataset (as it is the case from 2012-2015), the median is the average number of the two middle values. Thus, in the example of the CIR for Costa Rica, the two middle values are 80.49 and 81.21, which average 80.85. The same process was used for all calculations of medians.

I also question the validity of the comments on lines 156 to 159 since these trends are not statistically significant. In such cases, it would not be possible to state the existence of these trends.

Response: Trends can be observed, regardless of statistical significance. A 4-point time analysis is very sensitive to being skewed and it should be interpreted with caution. Please take a look at the caveat stated under Limitations: “Last, the limited number of four data pairs in the time series limits the sensitivity and power of the trend-over-time analyses.”

The discussions section brings elements not supported in the previous sections of the article (child pedestrian safety factors and Safe System Approach). This should not occur.

Response: Child pedestrian safety and Safe System Approach are a posteriori findings from our study, when we start the discussion section for findings that were new and unexpected. As stated in the introduction, there is a paucity of studies in the region, and those point out at CRS. However, our data reveal the MVC-related injuries point at pedestrian events. We have incorporated a sentence to link these two concepts.

The statement presented in the discussion that “Panama and Costa Rica has a trend to decrease mortality rates over time” is not statistically proven in the article.

Response: as stated before, trends may exist regardless of statistical significance. In compliance with transparent reporting guidelines, statistically significant trends are reported as such, while trends that do not reach statistical significance are simply reported as trends.

Reviewer 2 Report

Journal  IJERPH (ISSN 1660-4601)

Title: Epidemiological characteristics of road traffic injuries involving children in three Central American countries, 2012-2015

Manuscript ID

ijerph-1021965

Type Article

Number of Pages: 20

The article aims to analyse the epidemiological characteristics of road traffic injuries involving children in three Central American 3 countries, 2012-2015

Specifically, it was undertaken “A descriptive, retrospective study with publicly available data for Costa Rica, Panama, and Guatemala between 2012 and 2015”

Strengths:

The analysis is interesting

The article is written in an appropriate way.

Weakness:

1.-The study is not updated. Why do not provide a systematic analysis from 2012 to 2020. Are these data from 2016-2020 publicly available for Costa Rica, Panama, and Guatemala?  It would be great to know if the pattern of results has already changed or not, during more recent period.  Or perhaps include a wider number of countries in the analysis.

  1. -It must be justified the need of another publication about the same topic. There is no need to disseminate the same work. See for instance these previous/recent publications:

The requirements of originality or novelty are compromised. Do the results provide an advance in the current knowledge?

Comportamiento epidemiológico de la mortalidad de víctimas infantiles por accidentes de tránsito en tres países Centroamericanos durante los años 2012-2015

V Núñez Samudio

Centro de Investigaciones y Estudios de la Salud

2020

Traffic-Related Mortality and Fatality Among 0-to-14-Year-Olds in Three Central American Countries

V Núñez-Samudio, F Mayorga-Marín, I Landires

Available at SSRN 3396029

2019

  1. No adequate insights about the policies are provided at the discussion of the results. They require evaluation and intervention, talking about the significance of the work.

For instance see, Hazard Perception Training programs or evaluation of driver’s distractibility and ways to prevent it:

Assessment of proneness to distraction: English adaptation and validation of the Attention Related Driving Errors Scale (ARDES) and cross-cultural equivalence. Transportation Research Part F: 43, 357-365 DOI: 10.1016/j.trf.2016.09.004. 
Evaluating Smartphone-Based Virtual Reality to Improve Chinese Schoolchildren’s Pedestrian Safety: A Nonrandomized Trial, Journal of Pediatric Psychology, 2017 DOI: 10.1093/jpepsy/jsx147
Hazard Perception and Prediction test for walking, riding a bike and driving a car: “Understanding of the global traffic situation” PloS One,16, 2020    DOI: 10.1371/journal.pone.0238605
How are Distractibility and Hazard Prediction in driving related? Role of driving experience as moderating factor. Applied Ergonomics, 81, 102886, ISSN:0003-6870 DOI: 10.1016/j.apergo.2019.102886
The efficacy of a brief hazard perception interventional program for child bicyclists to improve perceptive standards  2018 Accident Analysis & Prevention 117 DOI: 10.1016/j.aap.2018.02.006

Author Response

Reviewer 2

Journal  IJERPH (ISSN 1660-4601)

Title: Epidemiological characteristics of road traffic injuries involving children in three Central American countries, 2012-2015

Manuscript ID ijerph-1021965         Type Article     Number of Pages: 20

 The article aims to analyse the epidemiological characteristics of road traffic injuries involving children in three Central American 3 countries, 2012-2015

Specifically, it was undertaken “A descriptive, retrospective study with publicly available data for Costa Rica, Panama, and Guatemala between 2012 and 2015”

Strengths:

The analysis is interesting

The article is written in an appropriate way.

Response: We would like to thank reviewer 2 for his excellent review work which has contributed significantly to improving our article. We are excited that reviewer 2 considers our study an interesting analysis and that it is written in an appropiate way.

Weakness:

1.-The study is not updated. Why do not provide a systematic analysis from 2012 to 2020. Are these data from 2016-2020 publicly available for Costa Rica, Panama, and Guatemala?  It would be great to know if the pattern of results has already changed or not, during more recent period.  Or perhaps include a wider number of countries in the analysis.

Response: Data published by local governments from 2016-2020 is not available in the sources used. Thus, updating these data for analytic and comparative purposes was not possible. Our initial attempt was to obtain data from all Central American countries, but only the three countries included had relatively recent data for our analytic purposes.

2.-It must be justified the need of another publication about the same topic. There is no need to disseminate the same work. See for instance these previous/recent publications:

The requirements of originality or novelty are compromised. Do the results provide an advance in the current knowledge?

Response:

This is an original research work. The subject area is within the research portfolio of two authors (IL and VNS), but does not duplicate previous publications.

For instance, the first work that reviewer 2 cites is actually a master's thesis by one of the authors of our article (VNS) that appears in the library repository of her university and has not been published previously in a peer-reviewed scholarly or scientific journal:  Comportamiento epidemiológico de la mortalidad de víctimas infantiles por accidentes de tránsito en tres países Centroamericanos durante los años 2012-2015. V Núñez Samudio. Centro de Investigaciones y Estudios de la Salud, 2020

The second paper that reviewer 2 cites is a preprint: Traffic-Related Mortality and Fatality Among 0-to-14-Year-Olds in Three Central American Countries. V Núñez-Samudio, F Mayorga-Marín, I Landires. Available at SSRN 3396029. 2019. In fact, a preprint is a full draft of a research paper that is shared publicly before it has been peer reviewed and published in a peer-reviewed scholarly or scientific journal (https://en.wikipedia.org/wiki/Preprint ).

With respect to references from previous work by our research group, Núñez-Samudio et al, (2020) [Reference 20] is a systematic review of public policies for child road safety. Núñez Samudio et al (2016) [Reference 6] is a prevalence study of child victims in MVCs in Panama and Spain, without reference to other Central American countries.

We consider this new body of data, analysis, and interpretation an original piece that gives context to the Central American region.

3. No adequate insights about the policies are provided at the discussion of the results. They require evaluation and intervention, talking about the significance of the work.

Response: We appreciate the extensive list of original publications. However, these focus mostly on interventions that translated into policies. In our study, we are generating baseline data analyses that hopefully will lay the ground to propose policy changes to prevent MVC-related injuries in children. That said, we have incorporated these policy implications in the Discussion.

Round 2

Reviewer 1 Report

The authors sought to meet the demands and complement the explanations in the text.
My main concern was with conclusions based on not statistically significant results.
However, the authors justify that it is usual in the study area (my background is in engineering). Therefore, if reviewers in the study area agree with the use of not statistically significant results, I am not opposed to accepting the paper for publication.

Author Response

We appreciate the opportunity to review and resubmit. We checked the Conclusions section and none state generalizations about trends (wether significant or not). As a matter of fact, the first paragraph referes to rates comparisons (not trends) among countries, the second paragraph refers to the connection of these rates to lack of specific legistlation that has been found in connection with MVCs, and the third paragraph is a call to action or future directions from our results.

Reviewer 2 Report

Please recognise at the limitation section that the study is not updated.

As you said: "Data published by local governments from 2016-2020 is not available in the sources used. Thus, updating these data for analytic and comparative purposes was not possible. Our initial attempt was to obtain data from all Central American countries, but only the three countries included had relatively recent data for our analytic purposes"

Author Response

We appreciate the insight of this limitation. Although data is not up-to-date, it is the best data that is publicly available. We have incorporated this limitation and caveat for results interpretation in the Limitations paragraph.